# Genomic Screening to Identify Food Trees Potentially Dispersed by Precolonial Indigenous Peoples

**DOI:** 10.3390/genes13030476

**Published:** 2022-03-08

**Authors:** Monica Fahey, Maurizio Rossetto, Emilie Ens, Andrew Ford

**Affiliations:** 1School of Natural Sciences, Faculty of Science and Engineering, Macquarie University, Sydney 2109, Australia; monica.fahey@hdr.mq.edu.au (M.F.); emilie.ens@mq.edu.au (E.E.); 2Research Centre for Ecosystem Resilience, Royal Botanic Gardens, Australian Institute of Botanical Sciences, Sydney 2000, Australia; 3Formerly Tropical Forest Research Centre, CSIRO Land and Water, Atherton 4882, Australia; andrewjford319@gmail.com

**Keywords:** ethnobotany, anthropogenic dispersal, propagule dispersal, insipient domestication, Indigenous, fruit size, chloroplast genome, non-crop species, rainforest assembly, genomic screening

## Abstract

Over millennia, Indigenous peoples have dispersed the propagules of non-crop plants through trade, seasonal migration or attending ceremonies; and potentially increased the geographic range or abundance of many food species around the world. Genomic data can be used to reconstruct these histories. However, it can be difficult to disentangle anthropogenic from non-anthropogenic dispersal in long-lived non-crop species. We developed a genomic workflow that can be used to screen out species that show patterns consistent with faunal dispersal or long-term isolation, and identify species that carry dispersal signals of putative human influence. We used genotyping-by-sequencing (DArTseq) and whole-plastid sequencing (SKIMseq) to identify nuclear and chloroplast Single Nucleotide Polymorphisms in east Australian rainforest trees (4 families, 7 genera, 15 species) with large (>30 mm) or small (<30 mm) edible fruit, either with or without a known history of use by Indigenous peoples. We employed standard population genetic analyses to test for four signals of dispersal using a limited and opportunistically acquired sample scheme. We expected different patterns for species that fall into one of three broadly described dispersal histories: (1) ongoing faunal dispersal, (2) post-megafauna isolation and (3) post-megafauna isolation followed by dispersal of putative human influence. We identified five large-fruited species that displayed strong population structure combined with signals of dispersal. We propose coalescent methods to investigate whether these genomic signals can be attributed to post-megafauna isolation and dispersal by Indigenous peoples.

## 1. Introduction

Historical plant dispersal by Indigenous peoples has been recorded in many different parts of the world and there is a growing recognition that ancient Indigenous populations had a significant influence on the composition and distribution of ecosystems [1,2,3,4,5,6,7,8]. However, the literature is sparse, due to a lack of published research, loss of cultural knowledge following colonisation, or because historical and academic observations have been blind to the diversity of Indigenous planting practices [9,10,11]. Contemporary Indigenous knowledge holders and colonial-era observations indicate that Indigenous groups from around the world have cultivated, traded and dispersed useful or culturally significant plants across the landscape [10,12,13,14,15,16,17]. Whether intentional or incidental, these plant translocation events would have expanded the abundance and geographic range of many species, and many populations of so-called “wild” species are therefore likely to represent living cultural artefacts (for example [9,18,19,20,21]).

Molecular studies have sought to reconstruct the demographic history of food plant species to investigate the origins and processes of domestication [22,23,24,25]. These studies generally focus on crops that have been extensively genotyped and for which different cultivars are well-described (for example [26,27,28,29,30]). However, little work has been carried out on the human-mediated dispersal of non-crop species (although see [31,32,33,34,35]). This could be in the form of “assisted migration” which is the movement of a species outside it’s natural range, “introduction” which is the establishment of new populations within a species’ existing range and “reinforcement” which is the planting of propagules from one population into another [36]. Retracing propagule dispersal by pre-colonial Indigenous peoples (hereon referred to as ‘Indigenous dispersal’) is important for cultural resource recognition and management and can yield insights to the long-term evolutionary impacts of translocations that can be applied to restoration activities.

To advance this field of study, we advocate for the use of simple genomic tests to screen for species that are likely to yield signals of Indigenous dispersal. While there are likely thousands of plant species around the world that have known or unknown histories of Indigenous cultivation, not all these can be successfully uncovered by genomic studies. Even in cases where there is abundant ethnographic evidence of translocation, many species will not carry an easy to interpret genomic signal. For instance, the quantity of propagules dispersed by historical human activities, and the distances over which they were dispersed, may not have been sufficient to create genetic structure that can be readily discriminated from that created by non-human modes of dispersal (as appears to be the case with *Camassia quamash*, [33]).

Our study focuses on the rainforests of eastern Australia. Contemporary Indigenous knowledge and early colonial records reveal extensive movement of various rainforest trees for food cultivation, ceremony and trading across the region [37,38]. However, the antiquity of these activities is not clear from the current state of the research. In a review of the subject, the Australian ethnographic literature is described as scant though detailed accounts of propagule movement, planting or cultivation, often with ceremonial elements [13]. The archaeological record suggests humans began to permanently occupy tropical rainforests in very low numbers at least 8 kya, with intensive settlement around 2 kya [39]. However, the archaeological record is also sparse, and it is possible that human rainforest occupation is much older. Additionally, the occurrence of pre-domesticates of New Guinea crops such as taro (*Colocasia esculenta*), yam (*Dioscorea alata*) and bananas (*Musa acuminata*) suggests that there was an “experimental horticultural province” [40] in northern Australia (including the northern section of the study region). These rainforest food plants spread to the Australian continent either while it was still contiguous with the New Guinea landmass in the terminal Pleistocene/early Holocene and/or via maritime human dispersal following sea-level rise in the mid-late Holocene [5,21].

Researchers that seek to retrace past Indigenous dispersal need to consider the dispersal capacity of faunal or other vectors in the study area. The relationship between fleshy fruit size and the dispersal potential of woody species has been successfully demonstrated among plants of the east Australian rainforests. Here, plants with small fleshy fruit are widely dispersed by birds and are typically characterised by low population structure and have potential for colonisation of new areas via long-distance dispersal (LDD) [41,42]. This genomic background would make it difficult to identify populations translocated by humans. In contrast, following the extinction of megafauna from the Australian continent between 50 to 16 kya [43], large-seeded plants lost an important mechanism for LDD and the ability to re-colonise areas of suitable habitat following the end of the Last Glacial Maximum (25–16 kya). Consequently, large-seeded rainforest species generally have greater between-population genomic divergence and occupy smaller geographic ranges than their small-fruited counterparts [42,44]. We anticipate that the strong population structure in large-fruited species would contrast with the genomic signal left by Indigenous dispersal events that post-date the megafauna extinction.

Here, we present a screening strategy that employs simple genomic tests to identify signals of dispersal within long-lived non-crop plant species that may be attributed to Indigenous peoples. We sought to investigate whether fleshy fruited species with a known history of Indigenous use carry genomic patterns that are distinctive from expected signatures of widespread faunal dispersal. We were also interested in whether these signatures could be identified in other species that are likely to have been a nutritious food source, but for which we lack historic evidence of their extensive use by Indigenous groups.

We employed an opportunistic sample design to develop nuclear DNA (nDNA) genotyping-by-sequencing and whole-chloroplast (cpDNA) SNP datasets of east Australian rainforest trees that fall into one of five fruit-trait categories that impact dispersal capacity. We tested for four genomic signals of dispersal with different patterns expected for species with a history of long-term isolation, long-term faunal-mediated dispersal, or dispersal following long-term isolation (Table 1, Materials and Methods 2.6). Signal 1 “low Fst values and the absence of isolation-by-distance (IBD)” is the outcome of recent and/or rapid dispersal, Signal 2 “admixture between sites” is produced by dispersal following long-term isolation, such as across a biogeographic barrier, Signal 3 “genomic outliers within sites” is produced by very recent long-distance dispersal (LDD) between formerly isolated sites. Signal 4 “long-distance dispersal of haplotypes” is produced by recent dispersal following long-term isolation and is distinct from range-wide haplotype sharing that is consistent with long-term faunal dispersal.

Species with signatures of dispersal following long-term isolation were regarded as candidates for further investigation of putative Indigenous dispersal histories. For these candidates, we outline a strategy to test specific dispersal hypotheses using more comprehensive sampling and coalescent analyses.

## 2. Materials and Methods

### 2.1. Study System

The study area extended along the coastal plains and ranges of eastern Australia from the tropical monsoonal rainforests of Iron Range (12°42′ S) in the Cape York Peninsula, Queensland (QLD) to the scattered subtropical forests around Glennifer, New South Wales (NSW; 30°22′ S; Figure 1). The study species are primarily located in the Australian Wet Tropics (AWT; 15°40′ to 19°15′ S) or northern NSW (NNSW) and some extend through the intervening regions of Central QLD (CQLD; ~20° to 24° S) and Southeast QLD (SEQ; ~25° to 28° S). There are several breaks in wet forest habitat within and between these regions [45].

During the Quaternary, climate-driven cycles of wet forest habitat contraction and re-expansion led to periods of genetic isolation and admixture for many rainforest species [41,46,47]. The AWT bioregion comprises a mosaic of tropical upland and lowland forests separated by drier corridors of mixed wet/dry habitats that act as “permeable” genetic or distributional barriers for some rainforest species [48,49,50,51,52]. This includes the Black Mountain Corridor (BMC) [49] and Cairns-Cardwell Lowlands (CCL) [44,48,50]. The subtropical rainforests in NNSW are highly fragmented, with upland sites isolated by extensive low-lying river systems. The Clarence River Corridor (CRC) is also a dry habitat break for some mesic species and has played a role in diversification between SEQ and upland regions of the mid-north coast of NSW [46,47,53].

Specialised large-fruit dispersers have been historically absent from NNSW and SEQ, and local dispersal rates are expected to be lower in the region [54,55]. Therefore, it is assumed that large fleshy fruit in southern forests have no means of long-distance dispersal except through human activity. This pattern appears to be less pronounced in the AWT [56], where fruit up to 62 mm can be locally dispersed (≤2 km) by non-volant vertebrates such as the southern Cassowary (*Casuarius casuarius johnsonii*) [57,58]. Meanwhile fruit bats (*Pteropus* spp.) and birds would facilitate dispersal of small-fruited species over longer distances across the whole study area.

### 2.2. Study Design

For our core analyses, we selected three groups of co-generic or closely related rainforest species with fleshy fruit and/or edible nutritious seed (Table 1). This includes 4 × *Elaeocarpus* (Elaeocarpaceae), 1 × *Pleioluma*, 1 × *Planchonella* and 2 × *Niemeyera* (Sapotaceae); 3 × *Endiandra* and 1 × *Bielschmiedia* (Lauraceae). The fruit of these species typically contain a single large seed that comprises most of the fruit. Additionally, we included *Castanospermum australe* (Fabaceae), in which the genomic impacts of dispersal by Bundjalung people in NNSW has been previously demonstrated [32]. For broader context, 12 additional species from other families were included in our initial analyses, 3 of which have inedible wind-dispersed fruit. We employed an opportunistic rather than comprehensive sample strategy that captured the core distribution of each of the study species, including their presence across putative biogeographic barriers (see Appendix A for details of sample collection).

We grouped species according to the following fruit traits: large fleshy and Indigenous used, small fleshy and Indigenous used, large fleshy, small fleshy, wind dispersed. Following [38], our fruit-size categories were based on maximum width and defined as large (≥30 mm) or small (≤30 mm; see Table 2). These categories correspond with the maximum size of fruit that can be ingested whole by the largest volant dispersers in the southern subtropical rainforests [57]. Fruit size was obtained from Australian Tropical Rainforest Plants Edition 8 (https://apps.lucidcentral.org/rainforest/text/intro/index.html accessed on 6 November 2021) or from plantNET (https://plantnet.rbgsyd.nsw.gov.au/ accessed on 6 November 2021). Species were categorised as Indigenous used if we found archaeological or ethnographic reports that indicate past or ongoing consumption by Indigenous groups in Australia (Table 2). The other species may also have been Indigenous used, but we could not find reports of this.

Among our study species, *Planchonella australis* is an anomaly since it has large fleshy fruit with 1–5 smaller seeds that can potentially be dispersed by fruit bats. Note that although fruit size is a variable trait, the lower end of the range is generally recorded from fruit with inviable seed or no seed at all and would not contribute to the gene pool of the species. Therefore, although a maximum fruit width of <30 mm has been recorded for *P. australis*, *Niemeyera whitei* and *Elaeocarpus johnsonii*, we included these in the large fruit categories as they are primarily much wider than 30 mm.

### 2.3. Simulation of Hypothetical Dispersal Scenarios

The premise of our screening strategy is that species with a history of post-megafauna isolation followed by recent Indigenous dispersal would produce genomic patterns that are distinct from widespread and long-term faunal dispersal. We sought to verify this assumption by simulating genetic differentiation of a species under 9 hypothetical dispersal scenarios. We calculated the pairwise Fst values from each of the simulated scenarios to determine whether patterns of population differentiation are identifiably distinct between long-term faunal and recent Indigenous dispersal (see Appendix B for full description of methods and results).

### 2.4. Nuclear and Chloroplast Genomic Methods

For all samples, nDNA extraction from leaf samples and SNP genotyping using DArTseq technology [75] was undertaken at Diversity Arrays Technology Pty Ltd. (Canberra, Australia). The DArTseq data were obtained from a prior study of *T. laurina* and *T. collina* [47] and *D. australis, C. serratifolia, Elaeocarpus reticulatus, N. dealbata* and *S. australis* (see Appendix A). The data for all other species are novel to this study. We followed [46] and filtered markers according to reproducibility average (proportion of technical replicates for which the marker score is consistent) and call rate (proportion of individuals with non-missing scores). We selected markers with a reproducibility average of at least 0.96 and a minimum call rate of 0.80.

In addition, we obtained novel comparative cpDNA sequence data for Sapotaceae and Lauraceae to determine the ancestral relationships between populations and samples. Whole-chloroplast sequencing was undertaken at Deakin Genomics Research and Discovery Facility (Geelong, Australia) and we assembled the genomes de novo with ORG.Asm [76]. We used CLC Genomics Workbench 20.0 (QIAGEN, Denmark) to inspect read quality and depth, and map reads against annotated reference sequences obtained from GeSeq [77]. We used the default settings to map Lauraceae samples against *E. globosa* (Accession: KT588614) and Sapotaceae samples against *Pouteria campechiana* (Accession: NC033501). For read conflicts, we used the quality score and vote options to determine the consensus sequence and we filtered variants with a coverage <8 or read consensus <60%. After removing areas of low coverage, the Lauraceae chloroplast sequence alignments ranged between 106,112 and 108,132 bp long (Table 2). The read coverage and quality were generally poorer for Sapotaceae species, and the cleaned alignments were between 84,279 and 87,841 bp.

We aligned the species libraries with the relevant reference sequence using the MEGA alignment function in Geneious Prime 2021.1.1 (Biomatters, New Zealand). To eliminate potential sequence errors, we removed non-synonymous variants in coding regions. To investigate the possibility of contamination in libraries with unexpectedly high variation, we extracted sequences that mapped to the *ycf1* and *ndhH* genes and used the BLAST function in GenBank with default settings to determine if any samples matched with libraries of algae or other distantly related species.

### 2.5. Assessment of Fruit Traits and Genetic Connectivity

We calculated pairwise genetic distances for all 25 species to verify our primary assumption that species with large fruit have lower dispersal rates than wind-dispersed or small-fruit species. Pairwise Fst values were calculated using the R package BEDASSLE [78] under the Weir and Hill model [79]. Then, for each species, we took the average of their pairwise Fst values at 50 km distance intervals, starting from 0 to 50 km up to 651 to 700 km. To visualise how fruit traits influence gene flow over each distance interval, we constructed violin plots of results organised by fruit trait. As there were only a few observations above 300 km, we plotted distance classes between 301–700 km together for visual clarity. Small-fruited species were expected to show lower pairwise Fst values than large fruit species.

### 2.6. Genomic Tests of Dispersal Signals

We sought to identify whether the core 15 study species show any of the four signals of putative Indigenous dispersal described in Table 1. For each signal, we expected different results for species with a history of long-term isolation, long-term faunal-mediated dispersal, or dispersal following long-term isolation. To identify candidates that warrant an investigation of Indigenous dispersal histories, we sought to eliminate species that show signals overwhelmingly consistent with long-term faunal-mediated dispersal or long-term isolation. Signals more consistent with dispersal following long-term isolation were hypothesised to be the outcome of Indigenous-mediated dispersal following isolation driven by the megafauna extinction.

To test for Signal 1 “combination of low Fst values and absence of IBD”, we performed a Pearson Mantel test on each species’ genetic and geographic distance matrices with 999 permutations (*p* = 0.05). The distance matrices were linearised Fst values (Fst/(1 − Fst)) against log geographic distance (km) and all calculations were made in the R package vegan 2.5–7 [80]. Where relevant, we subdivided the datasets to construct distance matrices within the AWT and NNSW. Given the sparse sample design and our aims to develop a screening strategy, we were more interested in identifying overall patterns of IBD than statistical significance.

Signal 1 is produced by recent and rapid radiation. This pattern may be attributed to extensive Indigenous dispersal, although other mechanisms of recent widespread migration cannot be excluded for small-fruited species. In the absence of Indigenous dispersal, large-fruited species were expected to show high Fst values consistent with long-term isolation. The impact of IBD was expected only in the absence of barriers. On the other hand, range-wide faunal dispersal in a stable system is likely to yield low Fst values in combination with IBD.

For Signals 2–3, we used the STRUCTURE-like genotype assignment algorithm implemented by R package sNMF [81] to assess the degree of shared ancestry between samples. We modeled *K* = 2–10 ancestral genotypes for each species, with 10 replicates per model. The cross-entropy criterion was used to evaluate model suitability in sNMF and we plotted the mean individual genotype assignments for *K =* 2–4 models. Given our sparse sample scheme may confound the genotype assignment algorithm, we verified the sNMF results with a principal components analysis (PCA) on the genomic variation among samples. Ordination was visualised in the first 3 primary axes of variation, with samples coloured according to latitude to determine whether genetic structure is geographic.

In our assessment of Signal 2 “admixture between sites”, we looked for sites where most samples had admixed sNMF profiles (e.g., ≤75% of the dominant genotype) in the optimal K model. Admixed profiles are a putative signal of secondary contact and admixture after many generations of isolation and could be facilitated by Indigenous dispersal or by faunal dispersal amongst small-fruited species. However, admixed profiles could alternatively be the outcome of incomplete lineage sorting due to vicariance, a recent bottleneck or admixture with an unsampled or extinct lineage [82]. We expected large-fruited species to show stronger population structure than small-fruited species, and Indigenous-used species to show some admixture of genotypes separated by barriers or disjunct regions. Regardless of human influence, small-fruited species were expected to show either a single genotype indicative of long-term range-wide connectivity or admixture consistent with post-glacial re-connectivity.

Signal 3 “within-site outliers” refers to samples that show a genotype that is distinct from most of the sample site (in the PCA and sNMF plot). Such a pattern may be produced by recent LDD and is hypothesised to be the outcome of recent Indigenous dispersal (reinforcement). Signals of LDD within small-fruited species may also be attributed to volant faunal dispersers, although this pattern is not expected.

To test Signal 4 “haplotype LDD”, the cleaned cpDNA alignments were exported for a Neighbour-Joining network analysis (epsilon = 0) of haplotypes in PopART (New Zealand) [83]. We looked for haplotype sharing or closely related haplotypes between otherwise highly genetically differentiated sites and/or disjunct sites as a putative signal of LDD. For large-fruited species, such a pattern is hypothesised to be the outcome of Indigenous-mediated reinforcement between previously isolated sites. Meanwhile assisted migrations or introductions may result in patterns consistent with rapid expansion, such as low haplotype variation between disjunct sites or a single widespread haplotype. Small-fruited species were expected to show extensive haplotype sharing and few mutations between haplotypes, indicative of long-term population connectivity. A single widespread haplotype may be indicative of rapid expansion facilitated by either Indigenous or faunal dispersal. Meanwhile large-fruited species without human influence were expected to retain strong haplotype differentiation between sites or across barriers.

## 3. Results

### 3.1. Fruit Traits and Genetic Connectivity

The violin plots of species-mean pairwise Fst shows that as a group, species with large fleshy fruit have higher median pairwise Fst values than the wind-dispersed or small fleshy-fruited species across all distance intervals excluding 201–250 km (Figure 2). This supports our founding premise that faunal vectors facilitate extensive gene flow within small fruit species, while large fruit species lack a mechanism of long-distance seed dispersal and thus have lower rates of gene flow. The large fruit though small-seeded *P. australis* has lower Fst values than the other categories and may be attributed to Indigenous-assisted dispersal or animal dispersal (Appendix A). Compared with fleshy fruit species, the range of Fst values is small in the wind-dispersed category and increases only marginally with geographic distance. This indicates that wind dispersal is relatively uniform in the study area, while gene flow within fleshy fruit species is sensitive to the type and/or availability of vertebrate dispersers.

### 3.2. Simulation Study

Overall, the nine simulated dispersal scenarios support the premise that long-term faunal dispersal and post-isolation Indigenous dispersal produce distinct patterns of genetic differentiation. The two hypothetical scenarios of post-glacial volant faunal dispersal show low Fst values though a prominent barrier effect (Figure A1 and Figure A2). As expected, the post-megafauna isolation model yielded the greatest population structure amongst all dispersal scenarios (Figure A3).

The Indigenous dispersal scenarios produced varying patterns of differentiation depending on the pattern of migration and the length of the migration period. For instance, the symmetric island model of migration in hd1 and hd2 (Figure A4 and Figure A5) yielded a greater homogenising effect than the distance-weighted migration of the faunal models. Models hd3 and hd6 with Indigenous dispersal 5000–4000 years ago exhibited higher Fst values due to the shorter and more ancient period of migration (Figure A6 and Figure A9). In contrast with all other models, the lack of migration combined with the range expansion in hd4 yielded high Fst estimates excluding between the two recently diverged deme0 and deme1 (Figure A7). The directional migration in hd5 and hd6 yielded higher Fst values and different population structure to the faunal dispersal scenarios (Figure A8 and Figure A9).

### 3.3. Genomic Tests of Dispersal Signals

We identified five candidates for the investigation of Indigenous dispersal that displayed at least two positive signals of dispersal: *C. australe,*
*E. insignis, B. bancroftii*, *E.*
*bancroftii* and *N. prunifera* (Table 3).

#### 3.3.1. Large Fruit with Known History of Indigenous Use

We assessed the northern and southern ranges of *C. australe* separately, due to the large geographic and genetic disjunction between the two regions. In the northern range, *C. australe* showed only one signal of dispersal (Table 3). We found low to moderate pairwise Fst values and a Mantel correlate consistent with IBD expected of non-anthropogenic dispersal (Table 2). The best supported sNMF models (*K =* 2–3) revealed divergence across the BMC and no outliers that may indicate LDD (Appendix A and Figure 3a(ii)). The PCA ordination was most concordant with *K =* 3, and both models suggest putative admixture or ILS across the BMC at sites ToS and CT (Signal 2; Figure 3a(iii)).

In the southern range, *C. australe* displayed two signals of dispersal that may be attributed to post-megafauna dispersal and presents a good candidate for further study (Table 3). First, we found low pairwise Fst values and a low Mantel correlate that suggests an absence of IBD consistent with recent or rapid migration (Signal 1; Table 2). The best-supported sNMF models assumed *K =* 2–4, though *K =* 4 was most consistent with the PCA ordination (Appendix A and Figure 3a(iv–v)). Both models show support for Signal 2, in which lowland sites (MP and Raz) have genomic profiles “admixed” between populations south of the CRC and upland sites north of the CRC. We did not find evidence of Signal 4 and the species shows unexpectedly strong structure between upland and lowland sites north of the CRC. This contrasts with the cpDNA results reported by Maurizio et al. (2017), which indicates widespread haplotype sharing in NNSW. The greater structure in the nDNA data may suggest that connectivity between sites has been lost in more recent generations.

*E. insignis* met one genomic signal of dispersal consistent with post-megafauna dispersal, though we identified it as a candidate for further study (Table 3). We did not find support for Signal 1, and *E. insignis* showed high Fst values, and a Mantel correlate consistent with IBD and limited faunal dispersal (Table 2). We did not find evidence of admixture between sites (Signal 2) and the best supported sNMF models (*K =* 1–2; Appendix A) revealed divergence across the CCL (Figure 3b(ii)). The PCA showed variation across the CCL and BMC (Figure 3b(iii)). We found evidence of LDD in the cpDNA data (Signal 4) that contrasts with the nDNA patterns. The haplotype network suggests dispersal across the CCL with a shared haplotype at sites B and CF that is highly differentiated from the other samples at those sites (Figure 3b(iv)). This pattern is more consistent with recent migration between the two sites rather than an ancestral haplotype. The lack of nDNA evidence for LDD may suggest dispersal has ceased in more recent generations, allowing for nDNA diversity to accumulate between sites.

*B. bancroftii* showed genomic patterns consistent with three putative signals of post-megafauna dispersal, making the species a good candidate for further study (Table 3). First, we found support for Signal 1 with a combination of low Fst values and Mantel correlate, that suggests an absence of IBD and recent or rapid migration (Table 2). The best supported sNMF models assume *K =* 1–2 (Appendix A) and together with the PCA reveal *B. bancroftii* is the only large-fruited species to show homogeneity among all sites excluding MtW (Figure 3c(ii–iii)). The PCA and sNMF also show one MtL sample clusters with MtW, potentially the outcome of LDD (Signal 3). The cpDNA data shows putative haplotype LDD (Signal 4) with one haplotype at MtL that is highly diverged from all others and may be a migrant from an unsampled population (Figure 3c(iv)). Alternatively, it may be a hybrid. Finally, the cpDNA network shows low variation within sites and high diversity between sites that suggests a long history of population isolation and bottlenecks. This is the opposite pattern to the nDNA data, suggesting that gene flow has shifted over time.

*P. australis* met only one signal of dispersal (Table 3) and is not considered a candidate for further study. We found low pairwise Fst values and a Mantel score that corresponds with IBD, consistent with long-term faunal connectivity (Table 2). The best supported sNMF model (*K =* 2; Appendix A) and the PCA show the primary source of variation is across the Wide Bay-Burnett region (WBB) in CQLD, and there is low variation albeit latitudinal structure between populations south of the barrier (Figure 3d(ii–iii)). One sample from CQLD has a genotype that clusters with the populations south of WBB suggesting past or recent LDD (Signal 3). The cpDNA network shows high variation consistent with vicariance across WBB and moderate haplotype diversity within and between the southern populations (Figure 3d(iv)). These patterns match that of the nDNA data and together suggest long-term population stability and periodic isolation rather than rapid migration and range expansion that we would expect of extensive Holocene faunal or anthropogenic dispersal. There is no cpDNA available for the CQLD sample that showed southern ancestry in the nDNA data, so it is unclear if the sample is a recent migrant.

*E. bancroftii* showed genomic patterns consistent with two signals of post-megafauna dispersal, and we considered it a candidate for further studies (Figure 1). We found support for Signal 1 with a combination of low pairwise Fst values and the absence of IBD, suggesting rapid migration (Table 3). The best supported sNMF model assumes *K =* 1 (Appendix A), and there is weak population structure in the PCA, primarily across the CCL (Figure 3e(ii–iii)). The *K =* 3 sNMF model is most concordant with the PCA and shows mixed genotypes that suggest admixture or ILS between sites within and north of the BMC (Signal 2. We did not have cpDNA data for this species, and so could not test for Signal 4.

#### 3.3.2. Large Fruit with Unknown Indigenous Use

We assessed *E. globosa* in the AWT and NNSW separately, due to the large geographic and genetic disjunction between the two regions. In the AWT, the genomic patterns in *E. globosa* were consistent with only one signal of dispersal (Figure 1). We found high pairwise Fst values over short distances that correspond with IBD, suggesting long-term isolation (Table 3). The best supported sNMF model (*K =* 3; Appendix A) and PCA revealed structure across the CCL, and outlier genotypes in WT potentially indicative of LDD (Signal 3; Figure 3f(ii–iii)). The cpDNA data are mostly concordant with the nDNA patterns and show haplotype divergence across the CCL and haplotype sharing between neighbouring sites (Figure 3f(iv)).

As there were only two *E. globosa* sites sampled in NNSW, we could not perform the Mantel or sNMF analyses for this region. We found *E. globosa* in NNSW matched one signal of dispersal and we did not consider it a candidate for further study (Figure 1). According to the PCA, most variation is between sites though there are outlier samples in HS, suggesting LDD (Signal 3; Figure 3f(iii)). The cpDNA shows the opposite trend to the nDNA data, with greater haplotype variation within BH and low variation between sites (Figure 3f(iv)). However, we did not find evidence of haplotype dispersal (Signal 4).

*E. compressa* did not show any genomic patterns consistent with dispersal and was not considered for further study (Table 3). We found high Fst values, and a high Mantel correlate consistent with limited faunal dispersal (Appendix A and Table 3). The primary source of variation was across the BMC according to the best supported sNMF model (*K =* 2; Appendix A) and PCA ordination, though there is some structure across the CCL (Figure 3g(iii)). The cpDNA network contrasts with the nDNA patterns and shows greater divergence between geographically proximate sites south of CCL while differentiation across the BMC is comparatively low (Figure 3g(iv).

The genomic patterns in *E. pubens* did not match any signals of dispersal and was not considered for further study (Figure 1). The populations in NNSW have moderate pairwise Fst values and a Mantel score consistent with IBD rather than rapid migration (Table 2). The best supported sNMF model (*K =* 2; Appendix A) and PCA reveal divergence between NNSW and CQLD (Figure 3h(ii–iii)). The PCA shows one outlier sample from NNSW, that may indicate LDD; however, this is not evident in the sNMF models. The cpDNA network conflicts with the nDNA data and shows greater variation within NNSW than between regions (Figure 3h(iv)).

We found only one genomic signal of dispersal in *N. prunifera* though we regard it as a candidate for investigation of Indigenous dispersal (Table 3). We did not find support for Signal 1 and *N. prunifera* has moderate pairwise Fst values and a high Mantel correlate that suggests IBD. The best supported sNMF model (*K =* 3; Appendix A) and PCA show differentiation between CQLD and the AWT and across the BMC, though no evidence of outliers or admixture (Figure 3i(ii–iii)). The cpDNA network displays high diversity within populations and only moderate differentiation between populations (Figure 3i(iv)). The relationships between some haplotypes are not geographically concordant and are consistent with LDD (Signal 4), including across the BMC. There is weaker population structure in the cpDNA compared with the nDNA data and may suggest past rapid migration followed by a decrease in dispersal over time.

*N. whitei* did not correspond with any signals and we did not consider it a candidate for further study (Table 3). We found high pairwise Fst values that correspond moderately with IBD, consistent with limited faunal dispersal (Appendix A; Table 2). The best supported sNMF model (*K =* 3; Appendix A) shows admixture or ILS, though this is not evident in the PCA clusters (Figure 3j(ii–iii)). The cpDNA network is concordant with the nDNA structure across the CRC, and the high variation suggests it is a long-term barrier (Figure 3j(iv)).

We found one signal of dispersal within *E. johnsonii* and did not consider it for further study (Table 3). The species has low to moderate Fst values, and a Mantel result consistent with IBD and long-term faunal dispersal (Table 2). The best supported sNMF model (*K =* 2; Appendix A) and PCA ordination show most variation is across the BMC and within sites (Figure 3k(ii–iii)). Both models indicate one MtSo sample has a mixed genotype that clusters with populations both sides of the BMC suggesting past LDD across the barrier (Signal 3). We did not have cpDNA data available to test for Signal 4.

#### 3.3.3. Small Fruit with Known History of Indigenous Use

*E. grandis* has genomic patterns that match two signals of dispersal, though we do not consider it a candidate for further study as we could not eliminate the influence of faunal vectors (Table 3). We performed separate Mantel tests for north and south of the BMC. To the south, we found low Fst values, and a low Mantel correlate consistent with rapid migration (Signal 1). North of the BMC, low Fst values combined with a moderate Mantel score consistent with IBD driven by widespread faunal dispersal (Appendix A and Table 2). The best supported sNMF model (*K =* 3; Appendix A) is concordant with the PCA (Figure 3l(ii–iii)). Both analyses identified three relatively homogeneous population clusters separated by the BMC and a 2° latitudinal disjunction to the south, consistent with extensive regional faunal dispersal. The models also show four samples from north of the BMC cluster with populations south of the barrier, potentially indicating LDD (Signal 3).

#### 3.3.4. Small Fruit with Unknown Indigenous Use

The genomic patterns in *E. discolor* align with two signals of dispersal; however, the species was not considered a candidate for further study as it showed patterns more consistent with widespread faunal dispersal (Table 3). We performed a Mantel test in NNSW only, as the other sites were too disjunct for a meaningful analysis. We found low pairwise Fst values and a very low Mantel score that suggests rapid migration consistent with Signal 1, though this is likely facilitated by widespread faunal dispersal (Appendix A and Table 2). Each of the sNMF models are equally supported and *K =* 4 shows admixed profiles between NNSW–SEQ and SEQ–CQLD, consistent with Signal 2 (Appendix A; Figure 3m(ii)). However, given SEQ and CQLD cluster separately in the PCA (Figure 3m(iii)), incomplete lineage sorting is more plausible than admixture. In the chloroplast haplotype network, *E. discolor* has one widespread haplotype distributed from the AWT to NNSW and some unique northern haplotypes differentiated along a latitudinal gradient (Figure 3m(iv)). This pattern may be attributed to Indigenous dispersal (Signal 4), though it is consistent with the nDNA data and more likely suggests periods of isolation across latitudinal barriers and subsequent widespread re-connectivity.

The patterns we found in *P. queenslandica* were consistent with long-term faunal dispersal and the species was not considered a candidate for further study (Table 3). We found low pairwise Fst values across more than 7° of latitude (Appendix A), though we had insufficient samples to test IBD within regions. The best supported sNMF models (*K =* 1–2; Appendix A) and PCA ordination show clinal variation in CQLD consistent with admixture or ILS between a northern and southern genotype (Signal 2; Figure 3m(ii–iii)). The cpDNA network shows range-wide haplotype sharing with moderate variation between haplotypes (Figure 3g). This may be attributed to Indigenous dispersal (Signal 4), though it is consistent with the nDNA data and more likely suggests a stable history of gene flow rather than LDD.

We did not find any genomic signals in *E. reticulatus* and it was not considered for further investigation of Indigenous dispersal (Table 3). We found high pairwise Fst values that moderately correlate with IBD (Table 2). The best supported sNMF model (*K =* 3; Appendix A) and the PCA ordination show the primary variation is across the CRC (Figure 3o(ii–iii). There is also variation between coastal and upland sites north of the barrier and between the sites west and south of the barrier.

## 4. Discussion

Reconstructing the demographic history of non-domesticated species with coalescent models can be a costly and challenging endeavour that requires extensive sampling and/or deep sequencing. Therefore, we sought to develop a simple and cost-effective screening strategy that can be used to screen out species with genomic patterns consistent with long-term widespread faunal dispersal and identify “candidate” species that show dispersal signals that warrant further investigation. The genomic signals we found in *C. australe* confirm the utility of our workflow, in which extensive Indigenous dispersal has already been demonstrated [32]. Our findings demonstrate that fast and widely used population genomic analyses can be employed to identify candidate species from opportunistically collected and somewhat sparse sample sets. Another advantage of our approach is that the genomic tests did not require any assumptions about the biogeographic history of the study species, making it a good first step. Our approach can be replicated in other study systems that have undergone a megafauna extinction and where Indigenous dispersal has been recorded.

We identified five candidates out of 15 species that show interesting dispersal patterns of putative Indigenous influence. Neither of the large-fruited study species displayed signals of ongoing or widespread dispersal. This raises the hypothesis that prior to putative Indigenous dispersal events within the candidate species, there was a considerable period of isolation driven by the megafauna extinction. As a next step, coalescent analyses can be used to estimate the antiquity of dispersal events. Based on the genomic patterns we found, we have suggested some hypothetical scenarios of past Indigenous dispersal to explore for each candidate (Table 4). Candidates can be co-analysed with ecologically similar and co-distributed species to contrast the influence of Indigenous versus faunal dispersal.

An important underpinning of our screening strategy was to eliminate faunal vectors (or other non-anthropogenic vectors) as the sole mode of dispersal within candidate species. To test the efficacy of our approach, we compared simulated and real genomic datasets of large- and small-fruited species with edible fruit. Most of the candidates we identified are large-fruited species with a known history of Indigenous use and carry signals of dispersal that are distinctive from widespread faunal dispersal. Likewise, the results of our simulation study demonstrate that long-term, range-wide faunal dispersal scenarios expected of small-fruited species yield patterns of population differentiation that are clearly distinct from species with a history of post-megafauna isolation followed by Indigenous dispersal.

Our findings confirm that dispersal-limited plants are more likely to carry genomic signatures that are suitable for investigating past Indigenous dispersal. First, we used pairwise genetic distance estimates to demonstrate that the large-fruited study species are more dispersal limited than the small-fruited and wind-dispersed species. Then, in the screening process, we found that the barrier effects evident in large-fruited species made distinctive signals of dispersal more apparent, particularly putative signals of LDD (Signal 3–4). Interestingly, we did not detect an overall trend of greater gene flow in the AWT (where a larger cohort of faunal dispersers still survives) compared with NNSW.

By the same token, we found that small-fruited species are generally less suitable for investigating Indigenous dispersal. Given their small fruit size, it is difficult to differentiate the relative influence of humans from volant frugivores or other natural dispersal vectors. For instance, widespread haplotype sharing in *P. queenslandica* and *E. discolor* may be attributed to Indigenous dispersal, though it is more consistent with faunal-mediated post-glacial recolonisation. Meanwhile, *Elaeocarpus grandis* and *P. australis* continue to be well-utilised by various Aboriginal groups and show nDNA signals of LDD. However, *P. australis* shows CpDNA structure more consistent with long-term isolation than extensive faunal- or Indigenous-mediated dispersal. In the case of *E. grandis*, rainforest restoration activities over the past few decades may also confound dispersal signals.

Out of all the candidates, *N. prunifera* is the only species for which we could not find any literature or verbal reports of use by Indigenous groups. The patterns we found for this species highlights the utility of genomic tests to investigate historical Indigenous dispersal, even in the absence of strong ethnographic evidence. On the other hand, we identified *E. globosa* as a poor candidate for Indigenous dispersal studies, despite archaeological evidence that the seed of morphologically similar and closely related Laurels were processed and consumed during the late Holocene [71,72]. It is worth noting that seed biology may prohibit successful attempts at long-distance dispersal of some food trees, as the seed of many Australian rainforest species do not store well and would not survive long journeys [84].

## 5. Conclusions

Overall, the workflow we have presented enabled us to identify genomic signals of dispersal that may be attributed to the past influence of Indigenous peoples and can be differentiated from widespread faunal dispersal. This includes species with edible fruit that lack published ethnographic evidence of Indigenous use. We found that the utilisation of both nDNA and cpDNA data was important for detecting putative dispersal signals, and its absence from the *Elaeocarpus* datasets made it more difficult to assess these species. We also found that three cpDNA samples per site was not always sufficient to identify dispersal events, and more samples would have aided interpretation where evidence of LDD was found in the nDNA. Therefore, we recommend that future screening studies utilise cpDNA sequence data for all samples.

## Figures and Tables

**Figure 1 genes-13-00476-f001:**
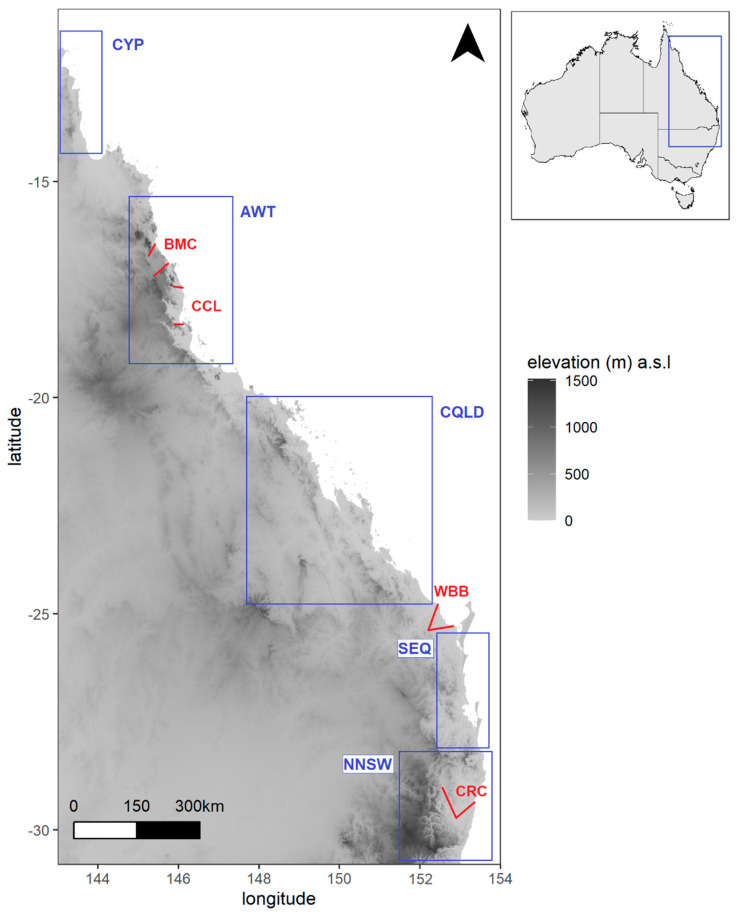
The study area in eastern Australia. Geographic regions separated by disjunctions of rainforest vegetation are indicated by the blue boxes. NNSW = Northern New South Wales, SEQ = Southeast Queensland, CQLD = Central Queensland, AWT = Australian Wet Tropics, CYP = Cape York Peninsula. Low elevation biogeographic barriers that structure the genomic variation in some of the study species are demarcated by red lines. CRC = Clarence River Corridor, WBB = Wide Bay-Burnett, CCL = Cairns-Cardwell Lowlands, BMC = Black Mountain Corridor.

**Figure 2 genes-13-00476-f002:**
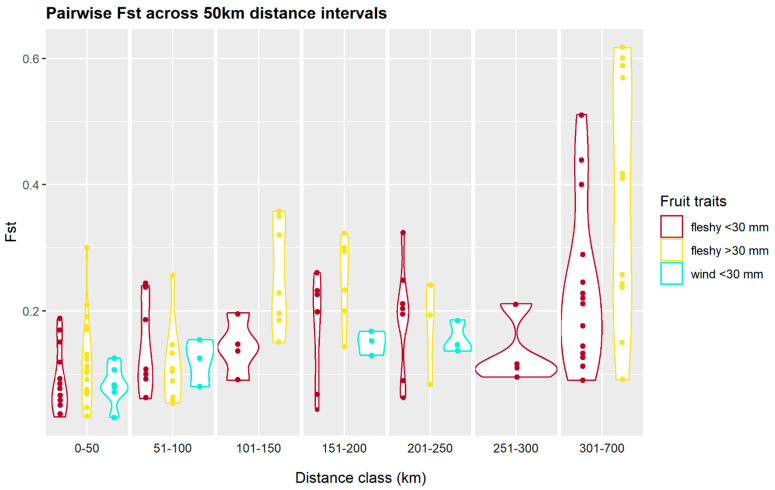
Violin plots of the average pairwise Fst values calculated for 25 species at 50 km distance intervals and coloured by fruit trait.

**Figure 3 genes-13-00476-f003:**
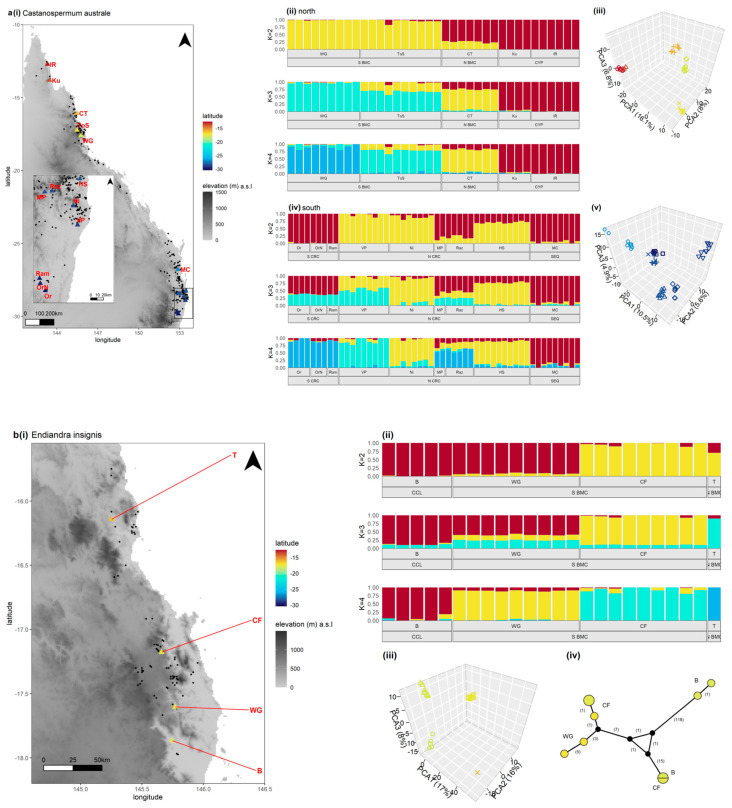
(**a**–**o**) The 15 study species evaluated for genomic signals of dispersal. For each species, (**i**) the distribution of the species in the study area is indicated by the black circles and the sample sites are coloured according to a latitudinal gradient defined by the extent of the study area. (**ii**) Genotype assignment proportions identified by sNMF, assuming *K* = 2–4. The sample site and geographic region (or position in relation to a barrier) are indicated by the bottom panel. (**iii**) Principal components analysis of nDNA genomic variance between samples, ordinated by first three primary axes of variation. Samples are coloured according to latitude and shape indicates sample site. (**iv**) Median-joining network of chloroplast haplotypes (epsilon = 0). Circles are proportional to the number of samples per haplotype and coloured by the latitude of the sample site. The number of mutations between haplotypes are in brackets, and the length of nodes are indicative but not directly proportional to number of mutations.

**Table 1 genes-13-00476-t001:** The patterns expected from four tests of dispersal assuming different dispersal traits and histories. For each signal, we expected different results for species with a history of long-term isolation, long-term faunal-mediated dispersal or dispersal following long-term isolation. Note that more than one dispersal scenario is hypothesised for species in the small fruit categories. Signal 1 = “low Fst values and absence of isolation-by-distance”. Signal 2 = “admixture between sites”. Signal 3 = “genomic outliers within sites”. Signal 4 = “haplotype long-distance dispersal”. **✓** = expected genomic signal from post-megafauna Indigenous dispersal. **✗** = genomic pattern not consistent with post-megafauna Indigenous dispersal. IBD = isolation-by-distance. LDD = long-distance dispersal.

Dispersal Trait	Signal 1	Signal 2	Signal 3	Signal 4
Small fruit faunal dispersed	**✗** Low Fst values & IBD (long-term dispersal)	**✓** Admixture between sites (dispersal following long-term isolation) **✗** Homogeneity among sites (long-term dispersal)	**✓** Within-site outliers (recent LDD) **✗** No outliers (long-term dispersal)	**✗** Range-wide haplotype sharing (long-term dispersal) **✓** Single widespread haplotype (recent rapid dispersal)
Small fruit Indigenous dispersed	**✓** Low Fst values & absence of IBD (recent rapid dispersal)	**✓** Admixture between sites (dispersal following long-term isolation) **✗** Homogeneity among sites (long-term dispersal)	**✓** Within-site outliers (dispersal following long-term isolation)	**✗** Range-wide haplotype sharing (long-term dispersal) **✓** Single widespread haplotype (recent rapid dispersal)
Large fruit faunal dispersed	**✗** High Fst values with or without IBD (long-term isolation)	**✗** Structure across barriers (long-term isolation)	**✗** Differentiation amongst sites and no outliers (long-term isolation)	**✗** Haplotype divergence (long-term isolation)
Large fruit Indigenous dispersed	**✓** Low Fst values & absence of IBD (recent rapid dispersal)	**✓** Admixture between sites (dispersal following long-term isolation)	**✓** Within-site outliers (dispersal following long-term isolation)	**✓** Haplotype sharing between differentiated sites (dispersal following long-term isolation) **✓** Single widespread haplotype (recent rapid dispersal)

**Table 2 genes-13-00476-t002:** The study species and their fruit traits, the genomic data used in the study and references that report use of each species by Indigenous Australians. Fruit traits: S = Small (<30 mm), L = Large (>30 mm), F = Fleshy, W = Woody, O = Other. Seed traits: L = Large, S = Small. nDNA = nuclear DNA. cpDNA = chloroplast DNA. Fst = Wright’s Fixation Index. Location: AWT = Australian Wet Tropics, CQLD = Central Queensland, SEQ = Southeast Queensland, NNSW = Northern New South Wales, SBMC = South of the Black Mountain Corridor in the AWT, NBMC = North of the Black Mountain Corridor in the AWT.

Family	Species	Common Names	Fruit Trait	Max. Fruit Width (mm)	Seed Number & Traits	nDNA Markers (SNPs)	cpDNA Sequence (bp)	Mantel score (*p* = 0.05) * Three Sites Only	Reported Indigenous Use
**Study Species**									
Fabaceae	*C. australe*	Moreton Bay chestnut, black bean, bean tree	LO	45	3–5 L	38,124 18,443 (north) 20,705 (south)		0.67 (*p* = 0.04) AWT 0.43 (*p* = 0.18) NNSW	‘Black bean was a staple food of many northern rainforest Aboriginal people and is still prepared and eaten today.’ (cited [59]). Ethnographic records of consumption by Indigenous peoples (AWT) [60,61,62,63,64]. Seed detoxification described in the AWT [65,66] and in NNSW/SEQ [32,67,68].
Lauraceae	*Bielschmiedia bancroftii*	Yellow walnut, yellow nut, Canary ash	LO	75 × 62	1 L	2080	108,132	0.36 (*p* = 0.33) AWT	Seed preparation described in the AWT [61]. Archaeological evidence of seed processing in the AWT [39,69].
Lauraceae	*Endiandra insignis*	Hairy walnut	LF	90 × 100	1 L	13,913	106,112	0.99 (*p* = 0.17) AWT *	Seed preparation described in the AWT [61]. Bush tucker guide (AWT) [70]. Archaeological evidence of seed processing (AWT) [69].
Sapotaceae	*P. australis*	Black apple, brush apple, wild plum, native plum	LF	50	1–5 S	24,873	86,899	0.63 (*p* = 0.17) NNSW *	Ethnographic records [67]. Bush tucker guide [70].
Elaeocarpaceae	*Elaeocarpus bancroftii*	Kuranda quandong, ebony heart, nutwood, Johnstone River almond	LF	55 × 40	1 L	17,085		0.14 (*p* = 0.32) AWT	Ethnographic records [67,71]. Bush tucker guide [68,70]. Archaeological records of seed preparation [72].
Lauraceae	*Endiandra compressa*		LF	71 × 60	1 L	4025	107,869	0.91 (*p* = 0.33) AWT	
Lauraceae	*Endiandra globosa*	Black walnut	LF	60 × 60	1 L	24,382	107,910	0.99 (*p* = 0.33) AWT	
Lauraceae	*Endiandra pubens*	Hairy walnut	LF	75 × 75	1 L	23,322	107,371	0.99 (*p* = 0.17) NNSW *	
Sapotaceae	*Niemeyera prunifera*		LF	50 × 50	1 L	22,778	84,279	0.91 (*p* = 0.12) AWT	
Sapotaceae	*N. whitei*		LF	20–50	1 L	10,669	87,841	0.61 (*p* = 0.33) NNSW *	
Elaeocarpaceae	*E. johnsonii*	Kuranda quandong	LO	40 × 25	1 L	1274		0.99 (*p* = 0.33) AWT *	Bush tucker guide described the seed as edible [73].
Elaeocarpaceae	*Elaeocarpus grandis*	Blue quandong, silver quandong, blue fig	SF	33 × 33	1 L	10,273		0.54 (*p* = 0.13) NBMC 0.13 (*p* = 0.35) SBMC 0.99 (*p* = 0.33) CQLD *	‘You can eat the thin layer of flesh of the ripe purple-blue fruits when flesh is soft.’ (cited [59]). Bush tucker guide describes edible fruit [70].
Elaeocarpaceae	*Elaeocarpus reticulatus*		SF	12 × 12	1 S	14,731		0.56 (*p* < 0.01) NNSW	B. McLeod describes the fruit as “good bush tucker tea” that can be eaten raw or as a jam [74] (NB: reference is from outside of study area).
Sapotaceae	*Pleioluma queenslandica*		SF	22 × 9	1 S	15,270	85,895		
Lauraceae	*Endiandra discolor*		SF	17 × 13	1 S	23,081	107,031	−0.05 (*p* = 0.42) NNSW	
**Fst Only**									
Lauraceae	*Endiandra introrsa*		LF	50 × 50		3461			
Lauraceae	*Bielschmiedia tooram*	Brown walnut, Tooram walnut	LF	55 × 35		3461			Bush tucker guide describes edible fruit [70]. Bush tucker guide describes edible seed [73].
Lauraceae	*Bielschmiedia volckii*		LF	67 × 65		3461			
Sapindaceae	*Diploglottis australis*	Native tamarind, tamarind tree, orange tamarind	SF	15		4640		0.88 (*p* < 0.01) NNSW	Ethnographic sources [60,67] and bush tucker guide [70] describe the culinary properties of the fruit.
Lauraceae	*Neolitsea dealbata*		SF	11 × 11		2881		0.91 (*p* < 0.01) NNSW	
Lauraceae	*Cryptocaria glaucesens*		SF	15 × 18		14,970		0.89 (*p* < 0.01) NNSW	
Elaeocarpaceae	*Sloanea australis*		SF	17 × 17		7429		0.59 (*p* < 0.01) NNSW	
Myrtaceae	*Tristaniopsis laurina*	Water gum, kanooka	W	10 × 6		13,841		0.59 (*p* < 0.01) NNSW	
Myrtaceae	*Tristaniopsis collina*	Mountain water gum	W	10 × 6		10,721		0.82 (*p* < 0.01) NNSW	
Cunoniaceae	*Ceratopetalum apetalum*	Coachwood	W	>8		659		0.75 (*p* < 0.01) NNSW	

**Table 3 genes-13-00476-t003:** Summary of dispersal signals found in the study species. The presence or absence of these signals can be used to evaluate whether a species would make a suitable candidate to investigate the influence of Indigenous dispersal. Signal 1 = “low Fst values and absence of IBD”. Signal 2 = “admixture between sites”. Signal 3 = “genomic outliers within sites”. Signal 4 = “haplotype LDD”. Species identified as candidates for Indigenous dispersal studies have an asterisk *.

Species	Fruit >30 mm	Verified Indigenous Use	Signal 1 nDNA	Signal 2 nDNA	Signal 3 nDNA	Signal 4 cpDNA
*C. australe* (CYP/AWT)	**✓**	**✓**	**✗**	**✓**	**✗**	**✗**
*C. australe* (SEQ/NNSW) *	**✓**	**✓**	**✓**	**✓**	**✗**	**✗**
*E. insignis* *	**✓**	**✓**	**✗**	**✗**	**✗**	**✓**
*B. bancroftii* *	**✓**	**✓**	**✓**	**✗**	**✓**	**✓**
*P. australis*	**✓**	**✓**	**✗**	**✗**	**✓**	**✗**
*E. bancroftii* *	**✓**	**✓**	**✓**	**✓**	**✗**	
*E. globosa* (AWT)	**✓**	**✗**	**✗**	**✗**	**✓**	**✗**
*E. globosa* (NNSW)	**✓**	**✗**			**✓**	**✗**
*E. compressa*	**✓**	**✗**	**✗**	**✗**	**✗**	**✗**
*E. pubens*	**✓**	**✗**	**✗**	**✗**	**✗**	**✗**
*N. prunifera* *	**✓**	**✗**	**✗**	**✗**	**✗**	**✓**
*N. whitei*	**✓**	**✗**	**✗**	**✗**	**✗**	**✗**
*E. johnsonii*	**✓**	**✗**	**✗**	**✗**	**✓**	
*E. grandis*	**✗**	**✓**	**✓**	**✗**	**✓**	
*E. discolor*	**✗**	**✗**	**✓**	**✓**	**✗**	**✓**
*P. queenslandica*	**✗**	**✗**	**✗**	**✓**	**✗**	**✓**
*E. reticulatus*	**✗**	**✗**	**✗**	**✗**	**✗**	

**Table 4 genes-13-00476-t004:** Candidate species that warrant investigation of historical Indigenous dispersal and suggested follow up studies. Species were identified as candidates if they displayed at least one of five genomic signals of dispersal that can be tested as anthropogenic vs. non-anthropogenic in future studies, and generated hypotheses on Indigenous dispersal scenarios. We considered species as weak candidates if they displayed genomic patterns from which putative Indigenous dispersal could not be differentiated from widespread faunal dispersal or if they showed an absence of dispersal events.

Species	Dispersal Hypotheses	Follow Up Studies
*C. australe*	(a) During the Holocene, *C. australe* was introduced to NNSW from a single northern lineage by humans or oceanic currents, and/or humans rapidly expanded its range in the region. (b) Extensive human-dispersal pathways in NNSW disrupted natural patterns of IBD evident in the north. (c) Upland populations in NNSW were established by humans. Founder effects and/or a subsequent lack of gene flow into these populations has led to drift.	(a) Sample upland sites and multiple lowland sites in multiple catchments across the species’ distribution, including CQLD. (b) Whole-genome sequencing for phased dataset that can be used to identify the geographic distribution of identity-by-descent blocks and recent coalescent events. Select population samples within each region to date the arrival of *C. australe* in NNSW and test for recent co-ancestry with northern genotypes. (c) Employ directional migration models between catchments to verify non-water modes of dispersal and test putative human-dispersal pathways inferred from ethnographic sources. (d) Employ directional migration models within catchments to verify that connectivity has been lost at upland sites.
*E. insignis*	(a) Mid-late Holocene human-mediated dispersal between two previously isolated sites, B and CF. (b) Holocene propagation along ancient walking routes between Atherton Tableland and the coast. (c) A subsequent decline or loss of dispersal has led to drift between populations.	(a) Sample additional populations at Atherton where there is archaeological evidence of *E. insignis* seed processing, and east along ancient walking routes between the Atherton Tableland and the coast. (b) To investigate dispersal across the BMC and between isolated upland sites, sample additional sites north of the BMC and at southern part of the range near the most differentiated population at site B. (c) Coalescent isolation with migration model to test for pre-Holocene vicariance between Bolinda and Curtain Fig, followed by Holocene-era LDD.
*B. bancroftii*	(a) Following megafauna decline, a long history of isolation has driven extreme haplotype differentiation between sites. Bottlenecks have reduced nDNA diversity and overall differentiation between sites. (b) Reinforcement—Holocene-era Indigenous dispersal facilitated limited migration between sites.	(a) Additional cp-sequencing per population to identify further evidence of dispersal events. (b) Isolation with migration coalescent models to test hypothesis of long-term vicariance followed by recent Indigenous-facilitated migration between sites.
*E. bancroftii*	(a) Rapid dispersal along cultural rather than geographic pathways. (b) Reinforcement—Holocene-era Indigenous dispersal facilitated limited migration and admixture across the BMC.	(a) Cp-sequencing to better infer dispersal between sites. (b) Coalescent model to evaluate ILS versus admixture between populations across the BMC.
*N. prunifera*	(a) Mid-late Holocene human-mediated LDD explains the disjunct distribution of *N. prunifera* in the AWT and CQLD and the migration of cp-haplotypes between geographically distant sites. (b) A subsequent decline or loss of dispersal has led to drift and strong nDNA structure.	(a) Sample additional populations in southern AWT to investigate the likelihood of vicariance versus LDD as the cause of disjunct distribution between AWT and CQLD. (b) Coalescent analysis to date divergence between AWT and CQLD. Divergence < 10 kya is likely human LDD, >21 kya is likely climate-driven vicariance. (c) Test for founder effects in CQLD, as support for LDD.

## Data Availability

The genomic datasets used in this study are available from the link (https://doi.org/10.5061/dryad.m0cfxpp5h) accessed on 2 March 2022.

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
