# Peer review of "Genomic Screening to Identify Food Trees Potentially Dispersed by Precolonial Indigenous Peoples"

_genes, 2022, doi:10.3390/genes13030476_

Round 1
Reviewer 1 Report
The authors propose a sequencing and data analysis strategy to explore whether Australian food tree species show population genetic patterns that may hint at dispersal by pre-colonial Indigenous people. The methods proposed in the manuscript will not lead to striking evidence for this but aim at identifying candidate species for further, more thorough analyses, which would perhaps involve explicit models for population demography with gene flow, e.g. with ABC methods. Then one could estimate model parameters such as time spans of gene flow events and analyse the range of possible values of these parameters (with confidence intervals or Bayesian sampling). Whether these demographic models are then in accordance with human-induced dispersal might perhaps be somewhat beyond statistical data analysis and rather a matter for the discussion part of a study.
The proposed workflow consists of certain sequencing methods, combining nuclear DNA with chloroplast DNA and searching for putative "signals" for dispersal by Indigenous people by applying common summary statistics and visualization methods from population genetics to these data. The authors describe possible scenarios of human-induced dispersal and argue how this would shift the summary statistics in a certain direction, but it is not clear how informative the summary statistics actually are, also because the descriptions of the scenarios are rather vague. Furthermore, the proposed workflow might be challenging to reproduce as it involves making subjective decisions (as the authors point out in the abstract in line 21). An analysis (e.g. by computer simulation) of sensitivity and selectivity of the proposed signals would be interesting, but this would require precise specifications of population-demographic models and depend on clear criteria and thresholds for the signals.
Major comments on some particular aspects:
The authors argue that it is unlikely that larger fruits have been dispersed in other ways than by humans after the Australian mega-fauna had gone extinct. They see this supported by the data as they obtain higher Fst for species with larger fruits than for those with smaller fruits. Even if this indicates that gene flow rates for larger fruits are smaller than for smaller fruits, it does not exclude the possibility that there still was some amount of gene flow, also not induced by humans. Furthermore, the authors focus on pre-colonial gene flow in there interpretations of the results and in the title of the manuscript, but if there is evidence for recent gene flow in the tree populations and even if the only explanation were human activities, would it really be possible to infer from the data whether this gene flow was due to pre-colonial or post-colonial human activities?
Regarding the interpretation of signal 4 (and also the comparison to nDNA and signal 5): As there is no recombination in cpDNA, the coalescent-based random variation is not averaged out on cpDNA. Therefore, one should be careful when drawing conclusions from population genetic analyses of cpDNA, in particular when comparing it to nDNA. A carful analysis is necessary, taking into account that the entire cpDNA of a species has evolved along a single coalescent tree.
If the work-flow is used to identify candidate species for more thorough downstream analyses, could this induce some ascertainment bias and how could this be compensated?
Minor comments:
l. 12: maybe split this sentence: "... histories however" -> "... histories. However"
l. 21: I would not put a comma after "process"
Perhaps some parts of section 2.1 can be moved to the intro.
l. 308: specie -> species
Throughout the manuscript the term "Indigenous people" is used, and I wonder whether "Indigenous peoples" may be more appropriate.
Author Response
The authors propose a sequencing and data analysis strategy to explore whether Australian food tree species show population genetic patterns that may hint at dispersal by pre-colonial Indigenous people. The methods proposed in the manuscript will not lead to striking evidence for this but aim at identifying candidate species for further, more thorough analyses, which would perhaps involve explicit models for population demography with gene flow, e.g. with ABC methods. Then one could estimate model parameters such as time spans of gene flow events and analyse the range of possible values of these parameters (with confidence intervals or Bayesian sampling). Whether these demographic models are then in accordance with human-induced dispersal might perhaps be somewhat beyond statistical data analysis and rather a matter for the discussion part of a study.
The proposed workflow consists of certain sequencing methods, combining nuclear DNA with chloroplast DNA and searching for putative "signals" for dispersal by Indigenous people by applying common summary statistics and visualization methods from population genetics to these data. The authors describe possible scenarios of human-induced dispersal and argue how this would shift the summary statistics in a certain direction, but it is not clear how informative the summary statistics actually are, also because the descriptions of the scenarios are rather vague. Furthermore, the proposed workflow might be challenging to reproduce as it involves making subjective decisions (as the authors point out in the abstract in line 21). An analysis (e.g. by computer simulation) of sensitivity and selectivity of the proposed signals would be interesting, but this would require precise specifications of population-demographic models and depend on clear criteria and thresholds for the signals.
Response: While we make subjective decisions to identify candidates, the tests we use in the workflow can be reproduced by other researchers. All phylogeographic analyses require subjective interpretation. The dispersal scenarios that we test for are intentionally non-specific or vague as we only aim to identify species that show genomic patterns against which dispersal signals may be detected. Specific dispersal scenarios are to be tested in the next phase.
Major comments on some particular aspects:
The authors argue that it is unlikely that larger fruits have been dispersed in other ways than by humans after the Australian mega-fauna had gone extinct. They see this supported by the data as they obtain higher Fst for species with larger fruits than for those with smaller fruits. Even if this indicates that gene flow rates for larger fruits are smaller than for smaller fruits, it does not exclude the possibility that there still was some amount of gene flow, also not induced by humans.
Response: Yes passive dispersal or local vertebrate dispersal can occur for large-fruited species post-megafauna extinction. Therefore in the introduction we mention “fruit up to 62 mm can be locally-dispersed (<2 km) by non-volant vertebrates such as the southern Cassowary (Casuarius casuarius johnsonii)” (line 128-130). This is why the signals we are testing for are long distance dispersal, which is very unlikely to occur without a faunal mode of dispersal. Hence we state “Therefore, it is assumed that large fleshy fruit in southern forests have no means of long-distance dispersal except through human activity” (line 132-133).
Furthermore, the authors focus on pre-colonial gene flow in there interpretations of the results and in the title of the manuscript, but if there is evidence for recent gene flow in the tree populations and even if the only explanation were human activities, would it really be possible to infer from the data whether this gene flow was due to pre-colonial or post-colonial human activities?
Response: The aim is to identify signals of dispersal, not to infer when the dispersal occurred. e.g. “Therefore, we sought to develop a simple and cost-effective screening strategy that can be used to identify “candidate” species that show dispersal signals that warrant further investigation”. (line 507-509). Dating the dispersal is proposed for the next stage of research “Based on the genomic patterns we found, we have suggested some hypothetical scenarios of past Indigenous dispersal to explore for each candidate (Table 3).” (lines 514-515). To clarify this, we added “As a next step, coalescent analyses can be used to estimate the antiquity of dispersal events.” (line 513).
Regarding the interpretation of signal 4 (and also the comparison to nDNA and signal 5): As there is no recombination in cpDNA, the coalescent-based random variation is not averaged out on cpDNA. Therefore, one should be careful when drawing conclusions from population genetic analyses of cpDNA, in particular when comparing it to nDNA. A carful analysis is necessary, taking into account that the entire cpDNA of a species has evolved along a single coalescent tree.
Response: Our interpretation of the cpDNA data is based on the fact that there is no recombination and hence it is a slow-evolving lineage. When comparing between the two datasets, we account for the fact that nDNA can accumulate diversity much more rapidly compared with cpDNA. As we understand this might be confusing, we have removed Signal 5 (contrasting patterns between genomes) and simply reported if there are different patterns in the results with interpretation if relevant. We made the following changes:
- Lines 327-329: “This contrasts with the cpDNA results reported by Maurizio et al. (2017), which indi-cates widespread haplotype sharing in NNSW, consistent with changed dispersal rates through time (Signal 5)” changed to “This contrasts with the cpDNA results reported by Maurizio et al. (2017), which indi-cates widespread haplotype sharing in NNSW. The greater structure in the nDNA data may suggest that connectivity between sites has been lost in more recent genera-tions.
- Added lines 344-346: “The greater connectivity evident in the cpDNA may suggest dispersal has ceased in more recent generations, allowing for nDNA diversity to accumulate between sites.”
If the work-flow is used to identify candidate species for more thorough downstream analyses, could this induce some ascertainment bias and how could this be compensated?
Response: I’m not sure I understand this comment. As you state the workflow is to identify candidate species for further study. Although interpretation of results found in the workflow could be used to generate hypotheses to test in the downstream analyses, a separate preliminary analysis does not bring any ascertainment bias.
Minor comments:
- 12: maybe split this sentence: "... histories however" -> "... histories. However" Response: change made
l. 21: I would not put a comma after "process" Response: change made
Perhaps some parts of section 2.1 can be moved to the intro. Response: We wanted the introduction to be short and to have a broader focus for an international focus, and hence have kept the Australia-specific background in section 2.1. If it reads better to move this to the introduction, we would be happy to make the change. - 308: specie -> species Response: change made
Throughout the manuscript the term "Indigenous people" is used, and I wonder whether "Indigenous peoples" may be more appropriate. Response: Agreed, change made.
Reviewer 2 Report
The paper brings valuable method for data processing and new interesting results in the field of human influence and dispersal interaction with food trees identified.
I have admit I have no concern about the manuscript and the study.
Author Response
There are no comments to respond to.
Round 2
Reviewer 1 Report
Thank you for your responses and the improvements in the manuscript. It is still hard to assess how informative the proposed methods are regarding gene flow mediated by Indigenous peoples. That is, will the proposed methods really be effective to detect candidate species? Especially because, as you clarify in your response, you aim to detect dispersal but not when it happened, it becomes even less clear to me how this could distinguish disposal by Indigenous peoples from earlier disposal at times when more large mammals were present of disposal at more colonial times. If this must all be done in the downstream analysis, this limits the benefit of the proposed steps to narrow down the set of candidate species. Regarding your question about ascertainment bias: The problem arises if statistical testing (or a somewhat equivalent Bayesian approach) is applied in the downstream analysis. If you first filter for species whose data show certain signals (or e.g. whose cpDNA coalescent show some features that may be random but could also be due to certain demographic effects) and then apply the downstream analysis only to the candidate species, this leads to a problem known as "significance chasing" or "p-hacking". Therefore some kind of compensation of the filtering may be needed in the downstream analysis, and it may be difficult to do this without losing too much statistical power.
Author Response
Thank you for your responses and the improvements in the manuscript. It is still hard to assess how informative the proposed methods are regarding gene flow mediated by Indigenous peoples. That is, will the proposed methods really be effective to detect candidate species? Especially because, as you clarify in your response, you aim to detect dispersal but not when it happened, it becomes even less clear to me how this could distinguish disposal by Indigenous peoples from earlier disposal at times when more large mammals were present of disposal at more colonial times. If this must all be done in the downstream analysis, this limits the benefit of the proposed steps to narrow down the set of candidate species.
Response: Thank you for the helpful feedback. It is clear from your review that our description of a genomic methodological workflow is distracting from the signals/results that we have found in the data. Therefore we have changed the angle of the manuscript and presented our data as a pilot study rather than a methodological template.
In response to your comment that our approach is undermined by the fact that we do not test for when dispersal events occurred - we have re-written the abstract, introduction and methods to make clear that we expect certain genomic patterns for species with a history of ongoing faunal dispersal (genetic connectivity), versus post-megafauna isolation (strong population structure) and post-megafauna dispersal (population structure combined with signals of dispersal) e.g lines 20-23, 120-122,289-291. We have included an additional table in the methods to summarise these expectations and how the four signals test for these three dispersal histories (Table 2). The description of the results have been slightly changed to reflect this, by reporting which of these dispersal histories are supported by the data e.g. lines 395-396, 408-409, 425-426, 454-455, 547-548, 556-557, 569-570, 582-583. Species that display patterns consistent with post-megafauna dispersal are regarded as suitable candidates for coalescent investigation to Indigenous dispersal.
In addition, we have followed your suggestion from the first review to move the description of the study system into the introduction (lines 73-88 and 97-102). We believe this helps set up the different genomic patterns we expect for the three dispersal scenarios that we test for.
The discussion was only marginally changed, to make emphasis on the results rather than the workflow. We do believe it is a useful pilot approach and provides a template that other researchers can follow. Hence we assert “Our approach can be replicated in other study systems that have undergone a megafauna extinction and where Indigenous dispersal has been recorded.” (lines 603-605).
We believe the analyses we have used are sufficient to support our interpretations of these three broadly described dispersal histories, and to raise the hypotheses of putative Indigenous dispersal that we outline in Table 4. The utility of our pilot study is demonstrated by the fact that we have reduced a list of 15 potential species to investigate down to 5. Although we do not date when dispersal events occurred, it is reasonable to hypothesise that a pattern of population structure combined with dispersal is the outcome of post-megafauna isolation followed by Indigenous-mediated dispersal. Hence we articulate, “Neither of the large-fruited study species displayed signals of ongoing or widespread dispersal. This raises the hypothesis that prior to putative Indigenous dispersal events within the candidate species, there was a considerable period of isolation driven by the megafauna extinction. As a next step, coalescent analyses can be used to estimate the antiquity of dispersal events.” (lines 611-614).
Regarding your question about ascertainment bias: The problem arises if statistical testing (or a somewhat equivalent Bayesian approach) is applied in the downstream analysis. If you first filter for species whose data show certain signals (or e.g. whose cpDNA coalescent show some features that may be random but could also be due to certain demographic effects) and then apply the downstream analysis only to the candidate species, this leads to a problem known as "significance chasing" or "p-hacking". Therefore some kind of compensation of the filtering may be